# Chemical composition and lipid profile of mare colostrum and milk of the quarter horse breed

Ícaro M. L. G. Barreto[1], Stela A. Urbano[1], Chiara A. A. Oliveira[2], Cláudia S. Macêdo[1‡], Luiz H. F. Borba[1‡], Bruna M. E. Chags[3‡]*, Adriano H. N. Rangel[1]

**1** Federal University of Rio Grande do Norte, Natal, RN, Brazil, **2** Federal University of Bahia, Salvador, BA, Brazil, **3** Federal University of Rio Grande do Norte, Natal, RN, Brazil

◎ These authors contributed equally to this work.
‡ These authors also contributed equally to this work.
* brunam.emerenciano@gmail.com

**Data Availability Statement:** The data set were uploaded to a public repository (Figshare) at the following DOI: 10.6084/m9.figshare.12578645.

## Abstract

The objective of this study was to characterize the chemical composition and lipid profile of colostrum and milk of purebred Quarter Horse mares. Thirty-four (34) purebred mares were selected, which were then separated into groups according to age, birth order and lactation stage. Colostrum samples were collected in the first six hours after delivery and milk samples from the 7th postpartum day, with intervals of 14 days until the end of lactation. The samples were refrigerated and sent to the Milk Laboratory of the University (*Laboleite—UFRN*), where they were analyzed for chemical composition. Colostrum was assessed by refractometry. The lipid profile was determined by gas chromatography through a separation of methyl esters. The data were tabulated and subjected to descriptive statistics and analysis of variance by the F-Test, and the groups were compared by the Tukey test using a significance level of 5%. There was high protein content and reduced lactose content for the colostrum of the Quarter Horse mares, differing from other breeds. The milk composition was not influenced by the mares' age. However, variations in the lactation stage and in the birth order of the Quarter Horse mares altered the milk's chemical composition. There is variation in the lipid composition of milk according to the lactation stage, without changing the characteristic profile of the mares' milk or diminishing the nutritional quality of the lipid fraction.

## Introduction

A considerable number of horses have been bred in several countries around the world in order to produce milk [1] because of its nutritional and therapeutic properties. According to Malacarne et al. [2] mares' milk is consumed by 30 million people worldwide, and has been studied as a milk substitute in newborns and pre-mature humans. Furthermore this milk can be used as a dietary supplement for older adults, recovering patients, and mainly children allergic to cow milk [1].

**Funding:** I declare that during this specific study we not received funds for financing this research. This work is part of the first author's Master Dissertation. The author Ícaro Marcell Lopes Gomes Barreto received a master's degree fellowship from Coordenação de Aperfeiçoamento de Pessoal de Nível Superior (CAPES).

**Competing interests:** The authors have declared that no competing interests exist.

On average, equine milk contains 6.5% lactose, 1.8% protein, 1.0% fat and 440 kcal/kg of energy [3]. It presents a desired protein profile in human food due to the whey:casein protein ratio and the spongiform structure of the micelles, which make it physiologically more digestible than cow milk [4]. The nutritional quality of the lipid fraction of equine milk is the result of small amounts of stearic and palmitic acids and large amounts of linoleic and linolenic acids [5], which has also supported indications for supplying equine milk to humans. Regarding colostrum, which has a dry matter much higher than milk (14 in milk to 29% in colostrum), it is important to highlight the high protein content (10%, on average), composed of immunoglobulins in 80% [6]. Colostrum fat is about 20% higher than milk fat produced in the initial third of lactation [7]. Bioactive peptide precursors, such as β-lactoglobulins and α-lactoalbumin, are present in colostrum from mares in considerable quantities [8]

Age, birth order, body weight of the mares, diet, environmental conditions and lactation stage have an influence on the chemical composition of milk [5, 9]. In addition to these factors, breed and genetics can change the composition of equine milk, especially the protein, fat and lactose levels [10]. Thus, the objective of this study was to characterize the chemical composition and lipid profile of colostrum and milk from purebred Quarter Horse mares of different ages, birth orders and lactation stages.

## Materials and methods

### Animal ethics and experimentation

The trial was submitted for evaluation by the Ethics Committee on the Use of Animals at *Federal University of Rio Grande do Norte* (protocol 062/2017), receiving approval registered under opinion number 058.062/2017. All animal management practices followed the recommendations of the National Council for the Control of Animal Experimentation (*CONCEA*) for the protection of animals used for animal experimentation and other scientific purposes, in accordance with the provisions of Law No. 11,794, of 8 October 2008, of Decree No. 6,899, of July 15, 2009.

### Experimental animals, material sampling and laboratory analysis

Thirty-four (34) purebred mares were selected from three different stud farms specialized in breeding and selection of Quarter Horses in Rio Grande do Norte state. The mares were separated into groups according to age (young: 3–5 years; adults: 6–10 years; seniors 11–19 years old), birth order (number of births throughout life, ranging from 1 to 6) and lactation stage (initial, medium and final third, considering a 180-day lactation)." All evaluated mares in this test gave birth naturally without cases of distocya In addition, mares which gave birth to a dead foals were discarded from the experimental group as well, with 34 mares remaining in the experimental group at the end of the selection. The collections took place between the months of July/2017 and September/2018, because the 34 births were not concentrated in a single period.

Colostrum samples were collected after parturition by manual milking not exceeding six hours after the event, and were obtained by manual milking after cleaning the udder and stored in previously sterilized plastic bottles. The composition (fat, protein, casein, lactose, total solids and defatted dry extract), Brix percentage by refractometry determination and lipid profile of the colostrum samples were analyzed.

Milk samples were collected from the 7[th] postpartum day with 14-day intervals, and thus continued until the end of lactation (180 days after parturition). The foal remained separated from the mare for two hours preceding the procedure on the day determined for collection in order to guarantee sufficient volume of milk for sampling accumulated in the udder. The

mares' udders were sanitized with a compress soaked with 70% alcohol, while the milker's hands were washed with clean water and neutral soap, dried with paper towels and also sanitized with 70% alcohol. The first three jets of milk were discarded at the time of collection, and then the cisterna udder was fully emptied by milking, with the milk being collected in a glass container previously sterilized in an autoclave.

The samples were identified, placed in an isothermal box containing artificial ice (4 to 8ºC) and sent to the *UFRN* Milk Quality Laboratory (*LABOLEITE*). The samples were subjected to electronic analysis by infrared absorption in DairySpec FT Bentley equipment to determine the chemical composition of the milk and colostrum. Qualitative analysis of colostrum was performed using a portable optical refractometer for sugar (Kasvi®, model K52-032, with a measurement range of 0 to 32% Brix and minimum division of 0.2%) after calibrating it with distilled water, as recommended by the manufacturer. One drop of colostrum was placed on the refractometer prism with the sample at room temperature and homogenized, and then reading was conducted through the monocular lens. The result in Brix% was obtained by the separation between the light area and the dark area formed on the equipment display after perpendicular disposition of the equipment to light.

The milk samples were lyophilized and the fatty acid methyl esters were obtained by adapting the methodology proposed by Kramer [11] to analyze the lipid profile. Approximately 0.8g of samples were weighed in glass tubes (16 x 150mm) with screw caps and septums in order to contain 15 to 30 mg of fat. Next, 2mL of hexane and 2mL of sodium methoxide (0.5M in methanol) were added to the tubes, followed by vortexing (30 seconds) and heating in a water bath (50˚C for 10 minutes). The tubes were subsequently cooled in running water and 3 ml of acetyl chloride (5% in methanol) were added to each tube, after which the tubes were heated again (80˚C for 10 minutes). Then, 1 ml of hexane and 10 ml of 6% $K_2CO_3$ were added, followed by vortexing for 1 minute and centrifugation (4,000 rpm for 2 minutes). The supernatant was transferred to 15mL Falcon tubes with approximately 1g of $Na_2SO_4$ mixture (previously oven dried) and activated carbon (1:1), followed by stirring (1 min.) and centrifugation (1 min. 4000 rpm). The supernatant was collected, transferred to an amber vial and then stored in a freezer at -20ºC.

The separation of the methyl esters from fatty acids was performed in a gas chromatograph (Focus GC—Thermo Scientific) equipped with flame ionization detector (CG-DIC) and SPTM-2560 capillary column (100m x 0.25mm x 0.20 μm—Supelco). The analysis parameters were: injector temperature of 250˚C; detector temperature of 280˚C; and 30:1 split ratio. The oven temperature was initially set at 140˚C, increasing at a heating rate of 1˚C/min to 220˚C; then remaining at that temperature for 25 minutes. Hydrogen gas was used as carrier gas at a flow rate of 1.5 mL/minute. The injections were performed in duplicates for each extraction and the injection volume was 1μL. The identification of the fatty acid methyl esters was performed by comparing the peak retention times of the samples with the retention time of the esters of the reference standard (GLC-674, Nu-Chek Prep, Inc.), and the result was obtained through normalizing the areas with the results expressed as a percentage.

The atherogenicity index (AI) and thrombogenicity index (TI) were calculated using the equation described by Ulbricht and Southgate (1991):

$$AI = (C12:0 + 4 \times C14:0 + C16:0)/[\Sigma MUFA + \Sigma(n-6) + \Sigma(n-3)]$$

$$TI = (C14:0 + C16:0 + C18:0)/[0.5 \times \Sigma MUFA + 0.5 \times \Sigma(n-6) + 3 \times \Sigma(n-3) + \Sigma(n-3)/\Sigma(n-6)]$$

## Statistical procedures

Data were tabulated in spreadsheets and subjected to descriptive statistics and analysis of vaance by the F-test. The groups were divided according to age, birth order and lactation stage, and were then compared using the Tukey test at a significance level of 5% for type I error. Only the different lactation stages were compared for the lipid profile analysis of milk. The statistical analyses were performed using the SAS (Statistical Analysis System) statistical package, and the analysis of variance was performed according to the following model:

$$Y_{ij} = \mu_i + group_j + residual_{ij}$$

In which:
$Y_{ij}$ = Dependent variables;
$\mu_i$ = Overall mean;
$group_j$ = Effect of the j[th] group (age, birth order and lactation stage) on dependent variables, being group 1 to 3;
$residual_{ij}$ = Residual effect.

## Results and discussion

The values obtained in the colostrum composition (Table 1) confirm its nutritional richness. It is important to highlight the high percentage of protein found for the colostrum of Quarter Horse mares (18%), which was higher than the average of 15% reported by Csapó et al. [12] in Hungarian Draught, Haflinger, Breton and Boulonnaise mares; and the 16% found by Pecka et al. [10] when they evaluated the colostrum of Arabian mares. The lactose content of the colostrum evaluated in this study (1.53%) also differed from that presented by other authors, such as: 3.4% cited by Salimei et al. [13]; 2.95% found by Pikul & Wójtowski [14]; and 2.46% presented by Pecka et al. [7]. The results suggest that the colostrum of Quarter Horse mares may contain more protein and be less dense in energy when compared to other breeds. However, the high fat content of this secretion stands out, being 2.7 times greater than the milk fat of the initial third of lactation and exceeding the parameter mentioned by Pecka et al. [7].

The ˚Brix values obtained were high, in line with the high protein content of the evaluated material, since approximately 80% of the colostrum protein corresponds to immunoglobulins [12]. The analyzed colostrum samples fall within the range of 20 to 30% of the refractive index established by Nath et al. [15], which classifies them as good, and represents an important factor for the passive transfer of immunity and consequently for establishing the newborn's health.

**Table 1. Chemical composition of colostrum and milk from purebred Quarter Horse mares.**

| Variable (%) | Colostrom | | Milk | |
| --- | --- | --- | --- | --- |
| | **Mean ± SD** | **CV (%)** | **Mean ± SD** | **CV (%)** |
| Fat | 1.70 ± 1.05 | 61.31 | 0.73 ± 0.45 | 61.30 |
| Total protein | 18.06 ± 2.00 | 11.09 | 1.68 ± 0.26 | 15.44 |
| Casein | 13.66 ± 2.00 | 14.63 | 1.26 ± 0.20 | 16.20 |
| Lactose | 1.53 ± 0.53 | 34.81 | 6.62 ± 0.30 | 4.45 |
| Total solids | 20.49 ± 2.36 | 11.53 | 10.00 ± 0.59 | 5.90 |
| DDE[1] | 19.95 ± 1.72 | 8.66 | 9.30 ± 0.27 | 2.89 |
| Brix% | 27.40 ± 4.15 | 15.18 | - | - |

[1]Defatted dry extract; CV = coefficient of variation; SD: standard deviation.

When analyzing the chemical characterization of milk (Table 1), a reversal between the protein and lactose concentrations is noticeable when the two secretions (colostrum and milk) are compared. However, as the lactose concentrations in milk are not as high as the protein concentration in colostrum, the levels of total solids and defatted dry milk extract are considerably lower than those observed for colostrum.

In studies conducted with Quarter Horse mares, Gibbs et al. [16] and Burns et al. [17] reported a variation of 1.8 to 2.9% for total milk protein, constituting values close to those found in this study. The lactose content of equine milk has previously reported to be higher than in other species [18] and the values obtained in the present study fall within the range reported in the literature for various horse breeds [7, 19–21], illustrating the importance of lactose as a source of carbohydrate in mares' milk [7].

The Fat content found in this study was lower than the range of 1.0–1.5% reported by Gibbs et al. [14] for Quarter Horse mares, and also below the average value of 1.25% reported by Salamon et al. [5], but higher than the 0.62% reported by Reis et al. [20] for milk from Mangalarga mares. Equine milk has low levels of fat when compared to milk from other species [2]; however, the measurement of this component in mares' milk is influenced by methodological details which are difficult to control and which translate into the high coefficient of variation (61.30%) presented in Table 1, and therefore deserve a brief discussion.

The small cistern of the mare's udder requires frequent milking and/or nursing by the foal throughout the day. Healthy foals nurse/drink several times an hour [22] and the ejection of milk requires the release of oxytocin [23]. When extrapolated to the sample collection methodology, these anatomical and physiological peculiarities reflect the difficulty of completely emptying the udder, which is directly related to the fat content of milk [16], since the residual fraction milk is rich in fat. Therefore, it is possible that the low fat content of milk found in this study is not solely and exclusively explained by genetic variations, but also because there is not enough oxytocin release during the sample collection to remove the residual milk fraction, resulting in in low-fat samples.

There was a significant effect of the lactation stage on the fat, total protein and casein levels, with the effect of such variations also occurring on the defatted dry extract levels (Table 2).

According to Markiewicz-Keszycka et al. [21], the evolution of lactation in mares leads to producing milk which is rich in lactose, but low in fat, protein and total solids. The fat levels at the end of lactation in this study were higher than those found at the beginning; Despite the tendency of increase observed for the fat content of milk according to the progress of lactation, it seems to be slightly lower than the 0.9% reported by Burns et al. [17] for milk from Quarter Horse mares at 150 days of lactation, thus reaffirming the difficulty of completely emptying the udder during collection and the permanence of alveolar milk in the mares evaluated in this study. Regarding protein contents and their fractions, there was a gradual reduction during

**Table 2. Effect of the lactation stage on the milk composition of purebred Quarter Horse mares (means ± standard deviation).**

| Variable (%) | Days in lactation | | |
|---|---|---|---|
| | **7–60** | **61–120** | **121–180** |
| Fat | 0.61 ± 0.41[b] | 0.85 ± 0.54[a] | 0.70 ± 0.36[ab] |
| Total protein | 1.94 ± 0.31[a] | 1.69 ± 0.18[b] | 1.53 ± 0.18[c] |
| Casein | 1.47 ± 0.24[a] | 1.27 ± 0.14[b] | 1.15 ± 0.15[c] |
| Lactose | 6.62 ± 0.23 | 6.60 ± 0.30 | 6.64 ± 0.32 |
| Total solids | 10.11 ± 0.50 | 10.09 ± 0.73 | 9.86 ± 0.46 |
| Defatted dry extract | 9.49 ± 0.26[a] | 9.29 ± 0.25[b] | 9.20 ± 0.24[b] |

[a, b, c] Different letters on the same line indicate statistical difference by the Tukey test (p<0.05).

lactation exactly as reported by Salimei and Fantuz [16]: a decrease of 20 to 25% of the total protein between the 28[th] and 150[th] days of lactation, accompanied by a 20 to 30% decrease of casein within the same period.

It is noteworthy that variations in the milk composition throughout lactation are essential for adjustments in the nutritional management of foals given the early development and rapid growth of horses at this stage of life (NRC, 1989). Add to this physiological nature, the expressive and particular muscular development of the Quarter Horse breed, which certainly requires an increase in protein intake so that there is no nutritional deficit or consequently losses in animal growth.

There was a difference (p<0.05) in the lactose levels in the effects of the birth order on the milk composition (Table 3), with a notable decrease in this component according to the maturity of the glandular tissue. Similar behavior was observed for defatted dry extract, most likely as a result of lactose variation, since this component is part of the dry extract.

It is important to consider that the physiological variations that occur in the mammary gland with the advancement of the longevity of the matrix can provide maximum performance with the maturity of the animal, changing the contents of some constituents [24, 25]. This can be an important detail in the management of herds that specialize in producing milk, or even in foal nutrition.

In evaluating the milk production and composition of primiparous and multiparous Quarter Horse mares, Pool-Anderson et al. [26] found greater production in multiparous mares, but did not report any variation in the milk secretion composition, differently from what occurred in this study. The fact that the lactose content is linked to the osmotic function and the milk production of the mammary gland [27] generated contrary expectations to the observed result, since young mares have lower production, which would lead to lower lactose content in the milk of low birth order mares.

Although low birth order mares are generally younger animals, there was no effect (p>0.05) of age on milk composition (Table 4).

Milk production in mammals increases with age until physiological maturity is reached [28], when there is a tendency to a functional reduction of the mammary gland caused by aging of the glandular tissue [29]. In addition, there is the dilution effect, in which the highest production tends to dilute the dry extract components [27], and so variation in the milk composition from mares of different ages was expected, especially between the evaluated extremes, but this did not occur in this study.

Tables 5 and 6 show the lipid composition of colostrum and milk from mares and the effect of the lactation stage on the lipid profile for milk, while Table 7 shows the results of relationships between fatty acids.

**Table 3. Effect of birth order on the milk composition of purebred Quarter Horse mares (means ± standard deviation).**

| Variable (%) | Birth order | | | | | |
|---|---|---|---|---|---|---|
| | 1 | 2 | 3 | 4 | 5 | 6 |
| Fat | 0.75 ± 0.52 | 0.73 ± 0.40 | 0.61 ± 0.42 | 0.84 ± 0.52 | 0.76 ± 0.39 | 0.76 ± 0.46 |
| Prot | 1.62 ± 0.14 | 1.62 ± 0.18 | 1.74 ± 0.33 | 1.61 ± 0.22 | 1.67 ± 0.23 | 1.74 ± 0.30 |
| Cas | 1.23 ± 0.12 | 1.22 ± 0.15 | 1.31 ± 0.24 | 1.21 ± 0.18 | 1.26 ± 0.19 | 1.30 ± 0.24 |
| Lact | 6.82 ± 0.24[a] | 6.70 ± 0.30[ab] | 6.67 ± 0.29 [abc] | 6.48 ± 0.27[c] | 6.52 ± 0.26[bc] | 6.59 ± 0.29[bc] |
| TS | 9.98 ± 0.89 | 10.09 ± 0.42 | 9.86 ± 0.68 | 10.06 ± 0.62 | 10.01 ± 0.43 | 10.02 ± 0.45 |
| DDE | 9.38 ± 0.16[a] | 9.36 ± 0.20[ab] | 9.34 ± 0.30[ab] | 9.19 ± 0.05[ab] | 9.26 ± 0.22[ab] | 9.27 ± 0.29[b] |

[a, b, c] Different letters on the same line indicate statistical difference by the Tukey test (p <0.05).

Fat = fat; Prot = total protein; Cas = casein; Lact = lactose; TS = total solids; DDE = defatted dry extract.

**Table 4. Effect of age on the milk composition of purebred Quarter Horse mares (means ± standard deviation).**

| Variable (%) | Age (years) | | |
| --- | --- | --- | --- |
| | **3–5** | **6–10** | **11–19** |
| Fat | 0.77 ± 0.43 | 0.71 ± 0.47 | 0.76 ± 0.41 |
| Total protein | 1.63 ± 0.11 | 1.67 ± 0.27 | 1.71 ± 0.31 |
| Casein | 1.23 ± 0.08 | 1.26 ± 0.21 | 1.29 ± 0.25 |
| Lactose | 6.64 ± 0.29 | 6.59 ± 0.30 | 6.66 ± 0.29 |
| Total solids | 9.99 ± 0.72 | 9.97 ± 0.61 | 10.04 ± 0.45 |
| Defatted dry extract | 9.32 ± 0.20 | 9.29 ± 0.29 | 9.29 ± 0.26 |

Saturated fatty acids prevailed over unsaturated faty acids in colostrum, with an emphasis on palmitic (C16:0), capric (C10:0), lauric (C12:0) and myristic (C14:0) acids, which increased the sum of saturated fatty acids (Table 5) and consequently the AGS:AGI ratio (Table 7). In the case of unsaturated acids, C18:1n9cis, C18:2n6cis and C18:3n3 stood out in comparison to the others. Similar behaviors for such acids have been reported by Pikul et al. [30] and Salamon et al. [5] for mares' colostrum; however, the values obtained in the present study were lower than those reported by these authors.

There was a higher concentration of unsaturated fatty acids in milk compared to colostrum regarding the lipid profile of mature milk when compared to colostrum; however, fatty acids (saturated and unsaturated) which stood out in the lipid profile of colostrum also did so in milk, with the values obtained herein within the range presented by Claeys et al. [31]. According to these authors, equine milk has higher proportions of unsaturated fatty acids when compared to milk from other species (especially cattle), due to the minimal occurrence of biohydrogenation before the absorption of unsaturated fatty acids.

There was an effect of the lactation stage on the lipid profile of mares' milk, with higher values of saturated fatty acids being obtained in the middle third of lactation. This is similar to the results presented by Orlandi et al. [32], especially in relation to C12:0, C14:0 and C16:0 acids. Pikul et al. [30] also found that the percentage of saturated acids decreased as lactation progressed. There was a significant reduction in linoleic acid in the final third of lactation, when investigating the most important unsaturated acids in our study, while linolenic acid concentrations were decreased in the middle of lactation. Consequently, the total amount of polyunsaturated fatty acids was also lower for the middle third of lactation.

**Table 5. Means and standard deviations of saturated fatty acids (peak area %) in Quarter Horse mares' colostrum and milk.**

| Fatty acid | Colostrom | Milk | Milk—days in milk | | |
| --- | --- | --- | --- | --- | --- |
| | | | **7–60 d** | **61–120 d** | **121–180 d** |
| C4:0 | 0.035±0.019 | 0.110±0.020 | 0.122±0.032[a] | 0.110±0.008[a] | 0.097±0.005[a] |
| C6:0 | 0.142±0.052 | 0.244±0.055 | 0.292±0.045[a] | 0.255±0.017[ab] | 0.185±0.288[bc] |
| C8:0 | 2.375±0.033 | 2.514±0.367 | 2.715±0.028[a] | 2.747±0.101[a] | 2.080±0.167[b] |
| C10:0 | 10.098±0.066 | 5.674±1.260 | 6.192±0.972[a] | 6.685±0.045[a] | 4.145±0.191[b] |
| C12:0 | 8.742±0.083 | 6.098±0.856 | 5.882±0.730[b] | 7.075±0.025[a] | 5.335±0.206[b] |
| C14:0 | 6.612±0.243 | 6.560±0.639 | 6.095±0.561[b] | 7.312±0.123[a] | 6.272±0.109[b] |
| C16:0 | 19.16±0.472 | 21.401±1.227 | 20.295±0.954[ab] | 22.437±0.162[a] | 21.485±1.229[ab] |
| C18:0 | 4.282±0.548 | 1.626±0.362 | 1.867±0.286[a] | 1.827±0.080[a] | 1.185±0.051[b] |
| ΣAGS | 53.652±1.364 | 46.491±3.818 | 45.322±3.632[b] | 50.795±0.522[a] | 43.357±0.628[b] |

[a,b,c] Means followed by different letters on the same line differ statistically from each other using the Tukey test at the 5% significance level. ΣAGS = sum of saturated fatty acids.

**Table 6. Means and standard deviations of unsaturated fatty acids (peak area %) in Quarter Horse mares' colostrum and milk.**

| Fatty acid | Colostrom | Milk | Milk—days in milk | | |
|---|---|---|---|---|---|
| | | | 7–60 d | 61–120 d | 121–180 d |
| C14:1 | 0.180±0.020 | 0.554±0.209 | 0.352±0.106[b] | 0.497±0.005[b] | 0.812±0.037[a] |
| C16:1 | 1.942±0.140 | 5.494±1.662 | 4.200±1.172[b] | 4.882±0.056[b] | 7.400±1.085[a] |
| C18:1n9cis | 15.700±0.186 | 16.688±0.971 | 16.64±1.405[a] | 16.600±0.367[a] | 16.822±1.144[a] |
| C18:2n6cis | 14.517±0.924 | 14.011±3.655 | 16.00±2.547[a] | 16.585±0.416[a] | 9.442±0.600[b] |
| C20:1n9 | 0.470±0.014 | 0.260±0.029 | 0.282±0.017[a] | 0.245±0.005[a] | 0.252±0.043[a] |
| C18:3n3 | 4.560±0.155 | 11.415±5.314 | 12.212±3.44[a] | 5.387±0.075[b] | 16.645±2.442[a] |
| C20:2n6 | 0.520±0.012 | 0.278±0.043 | 0.302±0.005[a] | 0.322±0.009[a] | 0.220±0.008[b] |
| C20:3n3 | 0.147±0.015 | 0.264±0.097 | 0.275±0.052[b] | 0.150±0.000[c] | 0.367±0.020[a] |
| ΣAGM | 18.697±0.295 | 23.099±2.547 | 21.547±2.636[a] | 22.310±0.390[a] | 25.440±2.312[a] |
| ΣAGPI | 19.980±1.041 | 26.105±3.236 | 28.985±0.985[a] | 22.560±0.443[b] | 26.770±2.971[a] |

[a,b] Means followed by different letters on the same line differ statistically from each other using the Tukey test at the 5% level of significance. ΣAGM = sum of monounsaturated fatty acids; ΣAGP = sum of polyunsaturated fatty acids.

The relationships between fatty acids presented in Table 7 show that even though some values obtained in this study are lower than those reported in the literature, there is a relevant nutritional advantage of the Quarter Horse mares' milk compared to milk of other breeds. The high concentrations of linoleic and linolenic acids, which have important biological functions, ensure that the proportions of fatty acids are nearly ideal [2].

Lower atherogenicity and thrombogenicity indices indicate the potential for atheroma and thrombus prevention [33] and were very similar to those presented by Pikul et al. [30] for Konik mares. The reduction in indices in the final third of lactation was also observed by Markiewicz-Kęszycka et al. [34].

## Conclusions

Quarter Horse mares produced colostrum with higher protein content and lower lactose content when compared to other horse breeds. The milk composition is not influenced by the mares' age; however, the lactation stage and the birth order alter the chemical composition of the milk of Quarter Horse mares. There is variation in the milk's lipid composition according to the lactation stage without changing the characteristic profile of mares' milk and without negative effects on the nutritional quality of the lipid fraction.

**Table 7. Means and standard deviations of the relationships between saturated and unsaturated fatty acids in colostrum and milk of Quarter Horse mares.**

| Fatty acid ratios | Colostrom | Milk | Milk—days in milk | | |
|---|---|---|---|---|---|
| | | | 7–60 d | 61–120 d | 121–180 d |
| Σn6 | 15.177±0.938 | 14.351±3.697 | 16.390±2.538[a] | 16.947±0.410[a] | 9.717±0.605[b] |
| Σn3 | 4.707±0.167 | 11.668±5.382 | 12.505±3.489[a] | 5.537±0.075[b] | 16.962±2.373[a] |
| n6:n3 | 3.225±0.155 | 1.692±1.122 | 1.437±0.603[b] | 3.062±0.065[a] | 0.577±0.044[b] |
| AGS:AGI | 1.390±0.080 | 0.954±0.153 | 0.902±0.136[b] | 1.130±0.0316[a] | 0.830±0.023[b] |
| AI | 1.415±0.085 | 1.105±0.173 | 1.010±0.150[b] | 1.312±0.034[a] | 0.995±0.040[b] |
| TI | 0.962±0.069 | 0.597±0.215 | 0.512±0.129[b] | 0.865±0.019[a] | 0.415±0.058[b] |

[a,b] Means followed by different letters on the same line differ statistically from each other using the Tukey test at the 5% level of significance. Σn6 = sum of omega 6 fatty acids; Σn3 = sum of omega 3 fatty acids; n6:n3 = ratio between omega 3 and omega 6 fatty acids; AGS:AGI = ratio between total saturated and unsaturated fatty acids; AI = atherogenicity index; TI = thrombogenicity index.

## Acknowledgments

The authors gratefully acknowledge the Laboratório de Qualidade do Leite from the Universidade Federal do Rio Grande do Norte for chemical analysis and the Haras Bom Pasto in Serrinha City, State of Rio Grande do Norte, Northeast of Brazil for making horses available.

## Author Contributions

**Conceptualization:** Stela A. Urbano, Chiara A. A. Oliveira, Luiz H. F. Borba, Bruna M. E. Chags, Adriano H. N. Rangel.

**Data curation:** Ícaro M. L. G. Barreto.

**Formal analysis:** Ícaro M. L. G. Barreto, Stela A. Urbano, Chiara A. A. Oliveira.

**Investigation:** Ícaro M. L. G. Barreto, Stela A. Urbano, Chiara A. A. Oliveira, Cláudia S. Macêdo, Adriano H. N. Rangel.

**Methodology:** Ícaro M. L. G. Barreto, Stela A. Urbano, Adriano H. N. Rangel.

**Supervision:** Stela A. Urbano, Chiara A. A. Oliveira, Cláudia S. Macêdo, Luiz H. F. Borba, Bruna M. E. Chags, Adriano H. N. Rangel.

**Validation:** Adriano H. N. Rangel.

**Visualization:** Adriano H. N. Rangel.

**Writing – original draft:** Ícaro M. L. G. Barreto, Stela A. Urbano, Chiara A. A. Oliveira, Cláudia S. Macêdo, Luiz H. F. Borba, Bruna M. E. Chags, Adriano H. N. Rangel.

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
