## [Decision Letter · Decision Letter 0]

19 May 2020

PONE-D-20-04990

Chemical composition and lipid profile of mare colostrum and milk of the quarter horse breed

PLOS ONE

Dear Dr. Chagas,

Thank you for submitting your manuscript to PLOS ONE. After careful consideration, we feel that it has merit but does not fully meet PLOS ONE’s publication criteria as it currently stands. Therefore, we invite you to submit a revised version of the manuscript that addresses the points raised during the review process.

Manuscript lacks in the quality of preparation. I agree with reviewers, and please consider the data presented in the manuscript, not support these conclusions; the sentences should be identified as hypotheses, review the referee comments and make your peer revision.

We would appreciate receiving your revised manuscript by Jul 03 2020 11:59PM. To enhance the reproducibility of your results, we recommend that if applicable you deposit your laboratory protocols in protocols.io, where a protocol can be assigned its own identifier (DOI) such that it can be cited independently in the future. For instructions see: http://journals.plos.org/plosone/s/submission-guidelines#loc-laboratory-protocols

We look forward to receiving your revised manuscript.

Kind regards,

Arda Yildirim, Ph.D.

Academic Editor

PLOS ONE

Journal Requirements:

2. Please amend your list of authors on the manuscript to ensure that each author is linked to an affiliation. Authors’ affiliations should reflect the institution where the work was done (if authors moved subsequently, you can also list the new affiliation stating “current affiliation:….” as necessary).

3. Please include a copy of Table 8 which you refer to in your text on line 269.

Additional Editor Comments (if provided):

There is a major flaw in the interpretation of the data, methodology, the statistical analysis, English language and redaction style. It is necessary to improve the manuscript by examining the questions that need to be clarified in a way. For your guidance, you can check the reviewers' comments. Thank you for giving us the opportunity to consider your work.

Reviewers' comments:

Reviewer's Responses to Questions

**Comments to the Author**

1. Is the manuscript technically sound, and do the data support the conclusions?

Reviewer #1: Partly

Reviewer #2: Yes

Reviewer #3: Yes

Reviewer #4: No

Reviewer #5: No

2. Has the statistical analysis been performed appropriately and rigorously? 

Reviewer #1: I Don't Know

Reviewer #2: Yes

Reviewer #3: Yes

Reviewer #4: No

Reviewer #5: No

3. Have the authors made all data underlying the findings in their manuscript fully available?

Reviewer #1: Yes

Reviewer #2: Yes

Reviewer #3: Yes

Reviewer #4: No

Reviewer #5: Yes

4. Is the manuscript presented in an intelligible fashion and written in standard English?

Reviewer #1: Yes

Reviewer #2: Yes

Reviewer #3: Yes

Reviewer #4: No

Reviewer #5: Yes

5. Review Comments to the Author

Reviewer #1: In fact, the authors characterized the chemical composition and lipid profile of colostrum and milk from purebred Quarter Horse mares. The authors introduced that the body weight of the mares, diet, environmental conditions have an influence on the chemical composition of mare’s milk (page 3, line 74). However, the study tested only the impact of age, birth order, and lactation stage effect on the milk composition.

Therefore, are the other factors (body weight, diet, environmental condition) unified for all the groups from the three different stud farms? This need to be clarified.

Moreover, the authors reviewed in the introduction the importance of the mare’s milk and its composition and there wasn’t any information on the research background on the mare’s colostrum even though it is tested in the study simultaneously with the mare's milk. Therefore, I think it’s better to introduce this as well.

Anyway, the authors reported important results that the Quarter Horse mare’s colostrum is of higher protein content and lower lactose content when compared to other breeds. The authors present that mares’ age does not impact the milk composition; however, the lactation stage and the birth order influence the chemical composition. The authors stated that the lactation stage alters the milk’s lipid composition without any changes in the milk profile or nutritional value.

On page 7, the authors discussed the results of the colostrum composition, where the fat value was not discussed even though it is presented in table 1, page 8.

On page 10 (lines 226-227) the authors stated ´´The fat levels at the end of lactation in this study were higher than those found at the beginning…..``, while in table 2 on page 10 the fat content was the highest in the middle lactation stage not the end stage?

On page 9 (lines 207-216) & on page 10 (lines 229-231) the authors argue that the reduction in the fat content in the mare’s milk is mainly attributed to the inability of the total evacuation of the mares udder, while in page 5 (line 109) they acknowledge that: ´´ the udder was fully milked…..``. This controversy need to be explained.

Reviewer #2: Dear authors,

The paper describes a fairly straight-forward (no disrespect intended) study on the composition of QH colostrum and milk, with a comparison to that of other horse breeds. The paper generally reads well, although specific suggestions have been made below.

A concern is that the reference numbers do not correspond to the appropriate references, which could indicate a lack of thorough checking of the manuscript before submission.

Specific comments/suggestions/questions:

Title: Quarter Horse with capital letters

41: subjected instead of submitted

43: reduced suggests a comparison – compared to what? (presumably other breeds)

68: I think contains would be better than is composed of

72: is the result of

72: large amounts rather than high amounts

76: breed rather than race

84: for instead of to

97: parturition rather than delivery

100: it was the Brix percentage that was analysed (not refractometry – that is the method of analysis)

103: parturition

104: for two hours, instead of during the last two hours

106: remove previously

109: collected instead of placed

128: added to each tube, after which the tubes were heated again….

151: subjected rather than submitted

155: analyses

167: obtained rather than verified

168: in instead of for

169: 18% or 18.1% would seem accurate enough (certainly when compared to whole percentages reported for other breeds)

169: (18.06%), which was higher than…

172: add the lactose content in the present study (in parentheses)

172: that instead of those

177: in line with instead of following

180: …as good, and represents an important factor for the passive transfer…

188: remove milk

189: However, as the lactose concentration in milk is not as high as the protein concentration in colostrum, the levels…..

194: The lactose content of equine milk has previously reported to be higher than in other species [16] and the values obtained in the present study fall within the range reported in the literature for various horse breeds [10, 17-19], illustrating the importance of lactose as a source….

199: remove ‘value regarding’ and lower instead of less

201: remove are

202: remove certainly

207: nursing by the foal instead of breastfeeding

208: healthy foals nurse/drink several times an hour

208: …and the ejection of milk requires the release of oxytocin.

211: consider ‘peculiarities’ instead of particularities and remove ‘on’

226: remove ‘a’

240: I’m not sure we can speak of ‘animal performance’ in foals

241: in instead of for

241: explain what you mean by ‘birth order’. Presumably this refers to the number of foals that the mare has had.

243: remove ‘breast’ (not a term used in horses)

246: table 3 – please include what the measure of variation is (presumable standard deviation)

I understand the lactose and DDE % in milk may differ statistically, depending on the number of foals, but I find it hard to believe that such a small difference would be (clinically/biologically) relevant and therefore question whether this should be discussed (at length). At least a comment could be added about any biological significance.

255: lower instead of smaller

264: In addition, there is the dilution effect, in which…

267: remove ‘fact’

276: the present instead of this (to make it clear to which study is being referred)

284: concentration instead of prevalence

284: …fatty acids in milk when compared….

287: range instead of threshold

288: remove ‘in fact’

The numbers of (many of the) references do not correspond to the correct papers. Please check and amend.

292: for instead of ‘of’

293: results

294: also found that the concentration (or percentage) of saturated acids decreased as lactation progressed. (Remove ‘in evaluating the lipid profile of mares’ milk throughout lactation in evaluating the lipid profile of mares’ milk throughout lactation.’)

296: …third of lactation, when investigating the most important…..

297: …linolenic acid concentrations were decreased….

298: Consequently, the total amount of…

301: Table instead of Tabela

307: reported instead of contained

308: species or breeds?

308: The high concentrations of linoleic and linolenic acids, which have important biological functions [2], ensure that the proportions of fatty acids are nearly ideal. Please provide reference for ideal values.

320: similar instead of close

326: other horse breeds

326: The milk composition is not influenced by…

330: ‘negative effects on the nutritional…’ instead of harming

335: horses

Reviewer #3: I can see that there has been a very good effort during this study. Here some advices to the authors:

I recommend an English review. There are some English mistakes that I have mentioned but there are other ones to be checked.

I understand that it is not easy to work with randomized conditions as mares were coming from different stables. This is why I consider that the composition of the food that they received should be added as an annex to cover all the possibilities of different compositions in colostrum and milk.

Were all the deliveries at term? If there were changes in the scheduled birth date (early delivery), the differences in the udder physiology may have produced alterations in the milk composition. I recommend to mention this but also to discuss it.

Lines 64-67: The phrase is incredibly long. Divide it to make easier the understanding.

Line 70: makes

Line 79 Birth order. What do you mean? Its position compared to the rest of individuals? The number of pregnancies?

Line 93. Were there any selection criteria?

Line 98: Colostrum samples were collected after delivery by manual milking.

Line 104: Erase last: If they were on the two hours preceding the procedure, It cannot be any others.

Line 105: In order to guarantee sufficient volume of milk

Results and discussion. I highly recommend to separate the results from the discussion. If results are separated from the discussion, the reader can obtain its own conclusions without being guided and then compare them with the manuscript.

Besides, I consider that this way results are very poorly presented

Lines 167-168. Separate them. It is a very strange sentence. I guess you mean: ...confirm its nutritional richness. It is important to highlight the high percentage...

Table 1: Colostrum instead colotrom. To be corrected in the other tables as well.

Line 189: I guess you mean that there are not high concentrations of lactose in milk. Rephrase

Line 199: Fat content found in this study...

References: Some references like 29 has not the abbreviation: Acta universitaria should be abbreviated to Acta univ. I did not check all the references.

Reviewer #4: Authors wanted to describe the composition of the quarter horse colostrum and milk. They seem to collect one sample of colostrum within 6h after birth and then, they collect milk samples, with no information about previous foal suckling, every 14 days. Analysis were performed on sample to give general data on milk (protein, lactose, caseins, lactose) and a detailed analysis of lipids.

General comments:

- The abstract should be totally re-written to explain the main method and results obtained in the study.

- They are few references and they are quite old: the newest one is from 2016 (only one) and all the others are older than 2014. Moreover, some links to reference are not accurate (example in line 287: reference 7 seems to be linked to a first author named Claeys and it’s not the case).

- Sometimes, English used in the paper is very strange, it should be reviewed by a native English speaker.

- The methods used and groups definition are poorly described:

o Until the results & discussion parts, it is impossible to know what are the group of ages, of birth orders, and of lactation stages are not defined by the authors. All along the paper, the size of the different groups is never mentioned. Statistical validity of this study is thus poor.

o Sampling of the colostrum is difficult to understand. What was sampled in each mare: Only one sample in each mare within 6 hours? Did the foal suckle before or after? Many questions that interfere with results.

o Some methods (example: infrared absorption in DairySpec FT Bentley) are poorly described, giving only the name of the manufacturer: rationale and descriptions are lacking.

o Statistical procedures are poorly described. Most of the time, it is impossible to understand what is compared with what. There is no information about normality of data, how this has been tested and the potential use of non-parametric test if the data don’t follow a Gaussian distribution. In the end, the general impression is that job has not been done properly.

- Results and Discussion

o The choice of the structure that associate results and discussion is inappropriate because it is already very difficult to understand the groups (not) defined in the methods section. Giving clear results would help to understand how samples and groups have been used through the study.

o Data observed are in quarter horses, but they may be similar in other horses: conclusion about the high protein level in quarter horse milk could be generalised to other breed if the same assays were used in other breeds.

o Frequently, it’s difficult to know if the results/discussion part is talking about colostrum, milk or both. It should be stated each time!

o Some statements are not statistically proven by the presented data or are not coming from references. We don’t care about what can be thought or imagined.

o In all presented tables, variables are expressed in % with no mention of percent of what: total milk, dry matter, defatted dry extract.

o Presentation of results is very poor, only giving tables that are difficult to read. Some graphs showing evolution or differences could help the reader.

o There is a major confusing factor that has not been statistically studied. It is obvious that mares with higher “birth order” (I would prefer using “lactation number”) will be older than primiparous one. A complete statistical model should have been realised to elude this confusion.

Conclusion

As the methods are poorly described and most of the time seem incomplete or inadequate, it is difficult to trust in the accuracy of the presented data. Thus, this paper is not suitable for publication.

Reviewer #5: Although the topic may be of interest,the manuscript lacks key elements:

The abstract contains unnecessary information; the keywords are not appropriate; the material and methods are not clear and lacking in information (eg. at 95 the authors do not explain how the groups were divided, they do not report the separation times of the foal for the colostrum; the amount of milk collected, at the time of samples, was not reported).

The statistical analysis should have consisted in ANOVA for repeated measurement and to include the effect of the farm. Furthermore, the two variables considered (age and parity (birth order) give similar information and one of the two variables should be removed from the model.

The references are not appropriated and are also dated (eg Cit 1, 2). In addition, eight tables are too numerous and the titles of the tables are not self-explanatory (eg Table 7).

Superfluous information is often reported in the text (e.g. l 106-110).

Finally, the results and conclusions are not satisfactory and should be reviewed in the light of the suggested considerations.

6. PLOS authors have the option to publish the peer review history of their article (what does this mean?). If published, this will include your full peer review and any attached files.

Reviewer #1: Yes: Manal Bakry Mohamed Hemida

Reviewer #2: Yes: Dr. Robin van den Boom, DVM, PhD, Dip.ECEIM

Reviewer #3: No

Reviewer #4: Yes: Jérôme Ponthier

Reviewer #5: No

---

## [Author Response · Author response to Decision Letter 0]

5 Jul 2020

The authors thank the Reviewers and Editor for the considerations. The research group has been working hard to get the work published in the journal.

Editor Requirements 06/24/2020

1. Thank you for updating your data availability statement. You note that your data are available within the Supporting Information files, but no such files have been included with your submission. At this time we ask that you please upload your minimal data set as a Supporting Information file, or to a public repository such as Figshare or Dryad.

Please also ensure that when you upload your file you include separate captions for your supplementary files at the end of your manuscript.

As soon as you confirm the location of the data underlying your findings, we will be able to proceed with the review of your submission.

The data set were uploaded to a public repository (Figshare) as follow DOI:

Digital Object Identifier

10.6084/m9.figshare.12578645

2. We note that you have included affiliation numbers 1,2 and 3 however only affiliations 1 and 2 have authors linked to them.

 The affiliation numbers were checked and corrected.

Editor Requirements:

Manuscript lacks in the quality of preparation. I agree with reviewers, and please consider the data presented in the manuscript, not support these conclusions; the sentences should be identified as hypotheses, review the referee comments and make your peer revision.

The manuscript has been reviewed as suggested by editor and reviewers. 

To enhance the reproducibility of your results, we recommend that if applicable you deposit your laboratory protocols in protocols.io, where a protocol can be assigned its own identifier (DOI) such that it can be cited independently in the future. For instructions see: http://journals.plos.org/plosone/s/submission-guidelines#loc-laboratory-protocols

This point has been analysed but it is not applicable.

Journal Requirements:

It has been checked as suggested

 2. Please amend your list of authors on the manuscript to ensure that each author is linked to an affiliation. Authors’ affiliations should reflect the institution where the work was done (if authors moved subsequently, you can also list the new affiliation stating “current affiliation:….” as necessary).

It has been checked as suggested

3. Please include a copy of Table 8 which you refer to in your text on line 269.

The number of tables has been reduced from 8 to 7.

Additional Editor Comments (if provided):

There is a major flaw in the interpretation of the data, methodology, the statistical analysis, English language and redaction style. It is necessary to improve the manuscript by examining the questions that need to be clarified in a way. For your guidance, you can check the reviewers' comments. Thank you for giving us the opportunity to consider your work. 

The manuscript has been reviewed as suggested by editor and reviewers. 

Reviewer #1: 

In fact, the authors characterized the chemical composition and lipid profile of colostrum and milk from purebred Quarter Horse mares. The authors introduced that the body weight of the mares, diet, environmental conditions have an influence on the chemical composition of mare’s milk (page 3, line 74). However, the study tested only the impact of age, birth order, and lactation stage effect on the milk composition.Therefore, are the other factors (body weight, diet, environmental condition) unified for all the groups from the three different stud farms? This need to be clarified.

The intention of introduction was to make it clear that there are other sources of variation in the quality of water milk. On this occasion, only the objectives of the study were investigated.

Moreover, the authors reviewed in the introduction the importance of the mare’s milk and its composition and there wasn’t any information on the research background on the mare’s colostrum even though it is tested in the study simultaneously with the mare's milk. Therefore, I think it’s better to introduce this as well.

The text was inserted (in the end of line 73) as follow: “Regarding colostrum, which has a dry matter much higher than milk (14 to 29%), it is important to highlight the high protein content (10%, on average), composed of 80% immunoglobulins (Csapó et al. 1995). Colostrum fat is about 20% higher than milk fat produced in the initial third of lactation (Pecka et al. 2012). Bioactive peptide precursors, such as β-lactoglobulins and α-lactoalbumin, are present in colostrum from mares in considerable quantities (Fessas et al. 2001). “

Anyway, the authors reported important results that the Quarter Horse mare’s colostrum is of higher protein content and lower lactose content when compared to other breeds. The authors present that mares’ age does not impact the milk composition; however, the lactation stage and the birth order influence the chemical composition. The authors stated that the lactation stage alters the milk’s lipid composition without any changes in the milk profile or nutritional value.

On page 7, the authors discussed the results of the colostrum composition, where the fat value was not discussed even though it is presented in table 1, page 8.

The text was inserted (Line 176) as follow: 

“However, the high fat content of this secretion stands out, being 2.7 times greater than the milk fat of the initial third of lactation and exceeding the parameter mentioned by Pecka et al. 2012.

On page 10 (lines 226-227) the authors stated ´´The fat levels at the end of lactation in this study were higher than those found at the beginning…..``, while in table 2 on page 10 the fat content was the highest in the middle lactation stage not the end stage?

The text has been rewritten (Page 19, Line 226 – 227) as follow:

"Despite the tendency of increase observed for the fat content of milk according to the progress of lactation, the values were still lower than the 0.9% reported by Burns et al. 1992"

On page 9 (lines 207-216) & on page 10 (lines 229-231) the authors argue that the reduction in the fat content in the mare’s milk is mainly attributed to the inability of the total evacuation of the mares udder, while in page 5 (line 109) they acknowledge that: ´´ the udder was fully milked…..``. This controversy need to be explained.

On page 9, lines 229-230, we refer to the difficulty of extracting the milk contained in the breast alveoli, which, as a rule, only happens in mares by suctioning the foal or applying oxytocin for removal by milking. In the methodology, when we mention total emptying of the udder, we speak of the udder cistern, of the milk already released from the alveoli.

In the methodology (on line 109), the text “and then the udder was fully milked” has been rewritten as follow:

“and then the cisterna udder was fully emptied by milking”. 

Reviewer #2: 

Dear authors,

The paper describes a fairly straight-forward (no disrespect intended) study on the composition of QH colostrum and milk, with a comparison to that of other horse breeds. The paper generally reads well, although specific suggestions have been made below.

A concern is that the reference numbers do not correspond to the appropriate references, which could indicate a lack of thorough checking of the manuscript before submission.

References and citations have been corrected 

Specific comments/suggestions/questions:

Title: Quarter Horse with capital letters

It was done

41: subjected instead of submitted

It was done

43: reduced suggests a comparison – compared to what? (presumably other breeds)

In the sbstract (line 43) the text has been rewritten) as follow:

 “There was a high protein and fat content, and low lactose for the colostrum of the Quarter Horse mares.”

68: I think contains would be better than is composed of

It was done

72: is the result of

It was done

72: large amounts rather than high amounts

It was done

76: breed rather than race

It was done

84: for instead of to

It was done

97: parturition rather than delivery

It was done

100: it was the Brix percentage that was analysed (not refractometry – that is the method of analysis) 

The refractometry have been corrected as follow:

“Brix percentage (refratometry determination)”

103: parturition

It was done

104: for two hours, instead of during the last two hours

It was done

106: remove previously

It was done

109: collected instead of placed

It was done

128: added to each tube, after which the tubes were heated again….

It was done

151: subjected rather than submitted

It was done

155: analyses

It was done

167: obtained rather than verified

It was done

168: in instead of for

It was done

169: 18% or 18.1% would seem accurate enough (certainly when compared to whole percentages reported for other breeds)

The value 18.04% was rounded to 18.1%.

169: (18.06%), which was higher than…

It was done

172: add the lactose content in the present study (in parentheses)

The lactose content “(1,53%) has been inserted on line 172 as suggested.

172: that instead of those

It was done

177: in line with instead of following

It was done

180: …as good, and represents an important factor for the passive transfer…

It was done

188: remove milk

It was done It was done

189: However, as the lactose concentration in milk is not as high as the protein concentration in colostrum, the levels…..

It was done

194: The lactose content of equine milk has previously reported to be higher than in other species [16] and the values obtained in the present study fall within the range reported in the literature for various horse breeds [10, 17-19], illustrating the importance of lactose as a source….

It was done

199: remove ‘value regarding’ and lower instead of less

It was done

201: remove are

It was done

202: remove certainly

It was done

207: nursing by the foal instead of breastfeeding

It was done

208: healthy foals nurse/drink several times an hour

It was done

208: …and the ejection of milk requires the release of oxytocin.

It was done

211: consider ‘peculiarities’ instead of particularities and remove ‘on’

It was done

226: remove ‘a’

It was done

240: I’m not sure we can speak of ‘animal performance’ in foals

 “Performance” has been replaced by “growth”.

241: in instead of for

It was done.

241: explain what you mean by ‘birth order’. Presumably this refers to the number of foals that the mare has had.

The description “birth order (number of births throughout life)” has been inserted on line 95.

243: remove ‘breast’ (not a term used in horses)

It was done.

246: table 3 – please include what the measure of variation is (presumable standard deviation)

The description “(means ± standard deviation)” has been included on caption of Table 2, 3 and 4.

I understand the lactose and DDE % in milk may differ statistically, depending on the number of foals, but I find it hard to believe that such a small difference would be (clinically/biologically) relevant and therefore question whether this should be discussed (at length). At least a comment could be added about any biological significance.

The following text has been included on line 250:

“It is important to consider that the physiological variations that occur in the mammary gland with the advancement of the longevity of the matrix can provide maximum performances with the animal's maturity, changing the contents of some constituents (Ribeiro et al., 2008; Rangel et al., 2008 ). ”

255: lower instead of smaller

It was done.

264: In addition, there is the dilution effect, in which…

It was done.

267: remove ‘fact’

It was done.

276: the present instead of this (to make it clear to which study is being referred)

It was done.

284: concentration instead of prevalence

It was done.

284: …fatty acids in milk when compared…. 

It was done.

287: range instead of threshold

It was done.

288: remove ‘in fact’

It was done.

The numbers of (many of the) references do not correspond to the correct papers. Please check and amend.

All references have been checked and They were corrigited when necessary. 

292: for instead of ‘of’

It was done.

293: results

It was done.

294: also found that the concentration (or percentage) of saturated acids decreased as lactation progressed. (Remove ‘in evaluating the lipid profile of mares’ milk throughout lactation in evaluating the lipid profile of mares’ milk throughout lactation.’)

It was done.

296: …third of lactation, when investigating the most important…..

It was done.

297: …linolenic acid concentrations were decreased….

It was done.

298: Consequently, the total amount of…

It was done.

301: Table instead of Tabela

It was done.

308: species or breeds?

It was done.

308: The high concentrations of linoleic and linolenic acids, which have important biological functions [2], ensure that the proportions of fatty acids are nearly ideal. Please provide reference for ideal values. 

The Reference “Malacarne et al. 2002” has been added as suggested.

320: similar instead of close

It was done

326: other horse breeds

It was done

326: The milk composition is not influenced by…

It was done

330: ‘negative effects on the nutritional…’ instead of harming

It was done

335: horses

It was done

Reviewer #3: I can see that there has been a very good effort during this study. Here some advices to the authors:

I recommend an English review. There are some English mistakes that I have mentioned but there are other ones to be checked.

I understand that it is not easy to work with randomized conditions as mares were coming from different stables. This is why I consider that the composition of the food that they received should be added as an annex to cover all the possibilities of different compositions in colostrum and milk.

Unfortunately this point is delicate for our research. The three stud farms that were willing to receive the University are involved in equestrian competitions and keep a considerable level of competitiveness among themselves. Considering the direct correlation between diet and performance, the owners did not provide information related to the nutritional management of the breeding stock.

Were all the deliveries at term? If there were changes in the scheduled birth date (early delivery), the differences in the udder physiology may have produced alterations in the milk composition. I recommend to mention this but also to discuss it.

The text “All evaluated mares in this test gave birth naturally without cases of dystopian parts. In addition, mares which gave birth to a dead foals were discarded from the experimental group as well” has been inserted on line 95 as suggested.

Lines 64-67: The phrase is incredibly long. Divide it to make easier the understanding.

The text (line 64-67) has been divided and writeen and as suggested

Line 70: makes

It was done

Line 79 Birth order. What do you mean? Its position compared to the rest of individuals? The number of pregnancies? 

Yes, the number of pregnancies. Here in Brazil, the transport of embryos has become frequent, as the number of foals produced by a matrix is often not the number of times that the mammary gland went into activity. So we evaluate this parameter

Line 93. Were there any selection criteria? 

The reproduction reports of the properties were monitored and matrices that would give birth within the experimental period were selected. Fort this study, just those mares which had no complications during pregnancy and delivery remained under evaluation

Line 98: Colostrum samples were collected after delivery by manual milking.

The term “by manual milking” has been inserted to text (Line 98) as suggested.

Line 104: Erase last: If they were on the two hours preceding the procedure, It cannot be any others.

“Last” was erased. 

Line 105: In order to guarantee sufficient volume of milk

“there being a” was erased.

Results and discussion. I highly recommend to separate the results from the discussion. If results are separated from the discussion, the reader can obtain its own conclusions without being guided and then compare them with the manuscript.

Besides, I consider that this way results are very poorly presented

Make the separation of the results from the discussion is not requirement from Plos one jornal so we decided keep the results and discussion together.

Lines 167-168. Separate them. It is a very strange sentence. I guess you mean: ...confirm its nutritional richness. It is important to highlight the high percentage...

It has been correct as suggested.

Table 1: Colostrum instead colotrom. To be corrected in the other tables as well.

Colotrom was replaced by colostrum.

Line 189: I guess you mean that there are not high concentrations of lactose in milk. Rephrase

It has been correct as suggested.

Line 199: Fat content found in this study...

It has been correct as suggested.

References: Some references like 29 has not the abbreviation: Acta universitaria should be abbreviated to Acta univ. I did not check all the references.

All references were checked and corrected when necessary.

Reviewer #4: Authors wanted to describe the composition of the quarter horse colostrum and milk. They seem to collect one sample of colostrum within 6h after birth and then, they collect milk samples, with no information about previous foal suckling, every 14 days. Analysis were performed on sample to give general data on milk (protein, lactose, caseins, lactose) and a detailed analysis of lipids.

General comments:

- The abstract should be totally re-written to explain the main method and results obtained in the study.

- They are few references and they are quite old: the newest one is from 2016 (only one) and all the others are older than 2014. Moreover, some links to reference are not accurate (example in line 287: reference 7 seems to be linked to a first author named Claeys and it’s not the case).

- Sometimes, English used in the paper is very strange, it should be reviewed by a native English speaker.

We appreciate the criterion with which our article was reviewed. The language and references have been checked and adjusted when needed as suggested.With regard to the bibliography consulted, for various reasons equine milk has not yet been extensively revised, which makes the number of articles on the subject limited. Even so, we can guarantee that our review included classic and important references on the topic, at the same time that we always try to cite the most recent studies as a way of showing the progress of science.

- The methods used and groups definition are poorly described:

The result tables show the well-characterized groups.

“(number of births in the matrix)” has been inserted on line 95 after birth order

Until the results & discussion parts, it is impossible to know what are the group of ages, of birth orders, and of lactation stages are not defined by the authors. All along the paper, the size of the different groups is never mentioned. Statistical validity of this study is thus poor.

o Sampling of the colostrum is difficult to understand. What was sampled in each mare: Only one sample in each mare within 6 hours? Did the foal suckle before or after? Many questions that interfere with results.

Yes. Each mare was evaluated only once in the postpartum period, given a small amount of colostrum and its importance for the newborn.

Some methods (example: infrared absorption in DairySpec FT Bentley) are poorly described, giving only the name of the manufacturer: rationale and descriptions are lacking.

Statistical procedures are poorly described. Most of the time, it is impossible to understand what is compared with what. There is no information about normality of data, how this has been tested and the potential use of non-parametric test if the data don’t follow a Gaussian distribution. In the end, the general impression is that job has not been done properly.

We are very sorry to have caused such an impression. The team worked hard throughout the research to expose the results so that it could contribute to the science.

- Results and Discussion

The choice of the structure that associate results and discussion is inappropriate because it is already very difficult to understand the groups (not) defined in the methods section. Giving clear results would help to understand how samples and groups have been used through the study.

Data observed are in quarter horses, but they may be similar in other horses: conclusion about the high protein level in quarter horse milk could be generalised to other breed if the same assays were used in other breeds.

Frequently, it’s difficult to know if the results/discussion part is talking about colostrum, milk or both. It should be stated each time!

The paragraphs that discuss colostrum and milk have been subtly flagged. Our intention was to produce a text with pleasant Reading.

Some statements are not statistically proven by the presented data or are not coming from references. We don’t care about what can be thought or imagined.In all presented tables, variables are expressed in % with no mention of percent of what: total milk, dry matter, defatted dry extract.

The percentages are for total milk. Such form of expression is recommended by the International Dairy Federation.

Presentation of results is very poor, only giving tables that are difficult to read. Some graphs showing evolution or differences could help the reader.

There is a major confusing factor that has not been statistically studied. It is obvious that mares with higher “birth order” (I would prefer using “lactation number”) will be older than primiparous one. A complete statistical model should have been realised to elude this confusion.

Here in Brazil it is not as obvious as it seems. If all mares started their reproductive life after puberty and had a birth each year, it would be as mentioned. However, here in Brazil equestrian sports strongly move the economy and many mares athletes start breeding only after retirement, while others, who were never athletes, start at three years of age. Thus, the wear and activity of the mammary gland are not similar in mares of the same age, which is why we decided to investigate the age and also the birth order of each matrix.

Conclusion

As the methods are poorly described and most of the time seem incomplete or inadequate, it is difficult to trust in the accuracy of the presented data. Thus, this paper is not suitable for publication.

Reviewer #5: Although the topic may be of interest,the manuscript lacks key elements:

The abstract contains unnecessary information; the keywords are not appropriate; the material and methods are not clear and lacking in information (eg. at 95 the authors do not explain how the groups were divided.

The text “...to age (young: 3 – 5 years; adults:6 – 10 years; seniors 11 – 19 years old), birth order (number of births throughout life, ranging from 1 to 6) and lactation stage (initial, medium and final third, considering a 180-day lactation).” has been added on line 95 in methodolgy section. 

They do not report the separation times of the foal for the colostrum; the amount of milk collected, at the time of samples, was not reported).

The statistical analysis should have consisted in ANOVA for repeated measurement and to include the effect of the farm. Furthermore, the two variables considered (age and parity (birth order) give similar information and one of the two variables should be removed from the model.

A similar question was raised by one of your peers, however, age and birth order do not provide similar answers.

If all mares started their reproductive life after puberty and had a birth each year, it would be as mentioned. However, here in Brazil equestrian sports strongly move the economy and many mares athletes start breeding only after retirement, while others, who were never athletes, start at three years of age. Thus, the wear and activity of the mammary gland are not similar in mares of the same age, which is why we decided to investigate the age and also the birth order of each matrix.

The references are not appropriated and are also dated (eg Cit 1, 2). In addition, eight tables are too numerous and the titles of the tables are not self-explanatory (eg Table 7).

Superfluous information is often reported in the text (e.g. l 106-110).

All references and tables has been reviewed as suggested. The number of tables has been reduced from 8 to 7.

Finally, the results and conclusions are not satisfactory and should be reviewed in the light of the suggested considerations.

---

## [Decision Letter · Decision Letter 1]

31 Jul 2020

PONE-D-20-04990R1

Chemical composition and lipid profile of mare colostrum and milk of the quarter horse breed

PLOS ONE

Dear Dr. Chagas,

Thank you for submitting your manuscript to PLOS ONE. After careful consideration, we feel that it has merit but does not fully meet PLOS ONE’s publication criteria as it currently stands. Therefore, we invite you to submit a revised version of the manuscript that addresses the points raised during the review process.

Needs to be checked native speaker English and all suggestions of reviewers should be addressed (even when you claim to have made changes). Please review the referee comments again and make your final revision.

We look forward to receiving your revised manuscript.

Kind regards,

Arda Yildirim, Ph.D.

Academic Editor

PLOS ONE

Additional Editor Comments (if provided):

Needs to be checked native speaker English and all suggestions of reviewers should be addressed (even when you claim to have made changes). Thanks

Reviewers' comments:

Reviewer's Responses to Questions

**Comments to the Author**

1. If the authors have adequately addressed your comments raised in a previous round of review and you feel that this manuscript is now acceptable for publication, you may indicate that here to bypass the “Comments to the Author” section, enter your conflict of interest statement in the “Confidential to Editor” section, and submit your "Accept" recommendation.

Reviewer #1: All comments have been addressed

Reviewer #2: (No Response)

Reviewer #3: All comments have been addressed

2. Is the manuscript technically sound, and do the data support the conclusions?

Reviewer #1: Yes

Reviewer #2: Partly

Reviewer #3: Yes

3. Has the statistical analysis been performed appropriately and rigorously? 

Reviewer #1: I Don't Know

Reviewer #2: I Don't Know

Reviewer #3: (No Response)

4. Have the authors made all data underlying the findings in their manuscript fully available?

Reviewer #1: Yes

Reviewer #2: Yes

Reviewer #3: Yes

5. Is the manuscript presented in an intelligible fashion and written in standard English?

Reviewer #1: Yes

Reviewer #2: No

Reviewer #3: Yes

6. Review Comments to the Author

Reviewer #1: (No Response)

Reviewer #2: Dear authors,

I feel the paper could still benefit from further English editing and the revision has introduced sections that are difficult to read/comprehend. I have included some specific comments below, but am somewhat concerned that quite a few of my suggestions have not been addressed (even when you claim to have made changes). You have also referenced a number of papers that I cannot access and/or read. I assume these are relevant and have been cited appropriately.

Specific comments/suggestions:

Title: Quarter Horse (captal Q and H)

43: just 'subjected' (not 'subjected instead of submitted')

45: the sentence has not been changed and I (still) think a comment should be added that the protein and lactose content are different to those (reported) in other breeds.

71: just 'contains' (not 'is contains')

75: remove 'composed of'

78: is it 14% in milk and 29% in colostrum, please clarify

91: 'submitted for' (rather than 'submitted to')

92: please explain UFRN

104: dystocia instead of dystopian parts

105: were these removed from the group of 34 mares, or never included? please clarify

106: presumably collections took place within two seasons within this time frame. Please clarify

180: 'in the colostrum' (rather than 'for the colostrum')

182: 'in Hungarian'

184: 'differed from that presented' (remove 'instead of those')

213: 'The fat content found in this study was lower...'

218: influenced or complicated rather than surrounded

222: 'nursing by the foal throughout'

242: I don't think it is correct to conclude that the fat% in your study is lower than the 0.9% reported by Burns, as both values will be a mean (with standard deviation) and you have not statistically compared the data. Maybe: seems to be slightly lower than the 0.9%.....

267-270: I was hoping for a statement to indicate that not all changes found are likely to be biologically relevant (because they are so small, even if they are statistically relevant). It is not clear (to me) what you are trying to say in this sentence.

298: 'the present study' (instead of 'this study')

304: 'concentration' (instead of 'prevalence')

304: '...fatty acids in milk compared to colostrum'

307: 'range' (instead of 'threshold')

308: remove 'in fact'

314: 'results' (instead of 'result')

350: 'The milk composition is not influenced by...'

359: 'horses' instead of 'horse'

Reviewer #3: Authors did correct most of the proposed comments/changes.

However, I still consider (as two of my peers) that results should be separated from the discussion as explained before. When there are several peers that are addressing this poorly presentation of results, the author should consider to change it even it is not a mandatory requirement of PlosOne as answered by the authors.

This is why, I let the decision of accepting the article with a gathered results/discussion section to the editors.

7. PLOS authors have the option to publish the peer review history of their article (what does this mean?). If published, this will include your full peer review and any attached files.

Reviewer #1: **Yes: **Manal Bakry Mohamed Hemida

Reviewer #2: No

Reviewer #3: No

---

## [Author Response · Author response to Decision Letter 1]

24 Aug 2020

The authors thank the Reviewers and Editor for the considerations. The research group has been working hard to get the work published in the journal. The manuscript has been reviewed as suggested by editor and reviewers. 

Editor Comments:

Needs to be checked native speaker English and all suggestions of reviewers should be addressed (even when you claim to have made changes). Thanks

The manuscript has been revided as suggested.

Reviewer #1: (No Response)

Reviewer #2: Dear authors,

I feel the paper could still benefit from further English editing and the revision has introduced sections that are difficult to read/comprehend. I have included some specific comments below, but am somewhat concerned that quite a few of my suggestions have not been addressed (even when you claim to have made changes). You have also referenced a number of papers that I cannot access and/or read. I assume these are relevant and have been cited appropriately.

The manuscript has been revided as suggested.

Specific comments/suggestions:

Title: Quarter Horse (captal Q and H)

The correction has been made.

43: just 'subjected' (not 'subjected instead of submitted')

The correction has been made.

45: the sentence has not been changed and I (still) think a comment should be added that the protein and lactose content are different to those (reported) in other breeds.

The correction has been made. 

In the lines 42-43 the text has been modified as follow:

“There was a high protein content and reduced lactose content for the colostrum of the Quarter Horse mares, differing from other breeds”

71: just 'contains' (not 'is contains')

The correction has been made. 

75: remove 'composed of'

The correction has been made. 

78: is it 14% in milk and 29% in colostrum, please clarify

In the lines 79-80 the text has been modified as follow:

Regarding colostrum, which has a dry matter much higher than milk (14 in milk to 29% in colostrum)...

91: 'submitted for' (rather than 'submitted to')

The correction has been made. 

92: please explain UFRN

The correction has been made. 

104: dystocia instead of dystopian parts

The correction has been made. 

105: were these removed from the group of 34 mares, or never included? please clarify

In the lines 110-112 the text has been modified as follow:

…In addition, mares, which gave birth to dead foals, were discarded from the experimental group as well, with 34 mares remaining in the experimental group at the end of the selection...

106: presumably collections took place within two seasons within this time frame. Please clarify

In the lines 112-114 the text has been modified as follow:

...The collections took place between the months of July/2017 and September/2018, because the 34 births were not concentrated in a single period...

180: 'in the colostrum' (rather than 'for the colostrum')

The correction has been made. 

182: 'in Hungarian'

The correction has been made. 

184: 'differed from that presented' (remove 'instead of those')

The correction has been made. 

213: 'The fat content found in this study was lower...'

The correction has been made. 

218: influenced or complicated rather than surrounded

The correction has been made. 

222: 'nursing by the foal throughout'

The correction has been made. 

242: I don't think it is correct to conclude that the fat% in your study is lower than the 0.9% reported by Burns, as both values will be a mean (with standard deviation) and you have not statistically compared the data. Maybe: seems to be slightly lower than the 0.9%.....

The correction has been made. 

The sentence has been rewritten in line 263 as follow:

… of lactation, it seems to be slightly lower than the 0.9% reported by Burns et al. [17]...

267-270: I was hoping for a statement to indicate that not all changes found are likely to be biologically relevant (because they are so small, even if they are statistically relevant). It is not clear (to me) what you are trying to say in this sentence.

The correction has been made. 

The sentence has been added in line 293 as follow:

...This can be an important detail in the management of herds that specialize in producing milk, or even in foal nutrition.

298: 'the present study' (instead of 'this study')

The correction has been made. 

304: 'concentration' (instead of 'prevalence')

The correction has been made. 

304: '...fatty acids in milk compared to colostrum'

The correction has been made. 

307: 'range' (instead of 'threshold')

The correction has been made. 

308: remove 'in fact'

The correction has been made. 

314: 'results' (instead of 'result')

The correction has been made. 

350: 'The milk composition is not influenced by...'

The correction has been made. 

359: 'horses' instead of 'horse'

The correction has been made. 

Reviewer #3: Authors did correct most of the proposed comments/changes.

However, I still consider (as two of my peers) that results should be separated from the discussion as explained before. When there are several peers that are addressing this poorly presentation of results, the author should consider to change it even it is not a mandatory requirement of PlosOne as answered by the authors.

This is why, I let the decision of accepting the article with a gathered results/discussion section to the editors.

The authors of this manuscript has been decided to keep the resultas and discussion together but it can modified as the Reviewer 2 suggested if it is a requirement of the Plos one journal.

---

## [Editor Report · Decision Letter 2]

27 Aug 2020

Chemical composition and lipid profile of mare colostrum and milk of the quarter horse breed

PONE-D-20-04990R2

Dear Dr. Chagas,

We’re pleased to inform you that your manuscript has been judged scientifically suitable for publication and will be formally accepted for publication once it meets all outstanding technical requirements.

Kind regards,

Arda Yildirim, Ph.D.

Academic Editor

PLOS ONE

Additional Editor Comments (optional):

Thanks for your hard work.
---

## [Editor Report · Acceptance letter]

3 Sep 2020

PONE-D-20-04990R2 

Chemical composition and lipid profile of mare colostrum and milk of the quarter horse breed 

Dear Dr. Chagas:

I'm pleased to inform you that your manuscript has been deemed suitable for publication in PLOS ONE. Congratulations! Your manuscript is now with our production department. 

Kind regards, 

on behalf of

Prof. Dr. Arda Yildirim 

Academic Editor

PLOS ONE